# Cascaded Adversarial Attack: Simultaneously Fooling Rain Removal and Semantic Segmentation Networks

## ABSTRACT

When applying high-level visual algorithms to rainy scenes, it is customary to preprocess the rainy images using low-level rain removal networks, followed by visual networks to achieve the desired objectives. Such a setting has never been explored by adversarial attack methods, which are only limited to attacking one kind of them. Considering the deficiency of multi-functional attacking strategies and the significance for open-world perception scenarios, we are the first to propose a Cascaded Adversarial Attack (CAA) setting, where the adversarial example can simultaneously attack different-level tasks, such as rain removal and semantic segmentation in an integrated system. Specifically, our attack on the rain removal network aims to preserve rain streaks in the output image, while for the semantic segmentation network, we employ powerful existing adversarial attack methods to induce misclassification of the image content. Importantly, CAA innovatively utilizes binary masks to effectively concentrate the aforementioned two significantly disparate perturbation distributions on the input image, enabling attacks on both networks. Additionally, we propose two variants of CAA, which minimize the differences between the two generated perturbations by introducing a carefully designed perturbation interaction mechanism, resulting in enhanced attack performance. Extensive experiments validate the effectiveness of our methods, demonstrating their superior ability to significantly degrade the performance of the downstream task compared to methods that solely attack a single network.

## CCS CONCEPTS

• **Computing methodologies → Computer vision tasks**; • **Security and privacy → Software and application security**.

## KEYWORDS

Adversarial Attack, Adverse Weather, Semantic Segmentation, Rainy Scenarios

## 1 INTRODUCTION

With the rapid development of digital media and deep neural networks (DNNs), various computer vision algorithms have gained significant attention for real-world applications such as autonomous driving [6, 37, 39] and video surveillance [3, 33]. However, the presence of rainy weather, a common atmospheric condition, poses

*ACM MM, 2024, Melbourne, Australia*
© 2024 Copyright held by the owner/author(s). Publication rights licensed to ACM.
ACM ISBN 978-x-xxxx-xxxx-x/YY/MM
https://doi.org/10.1145/nnnnnnn.nnnnnnn

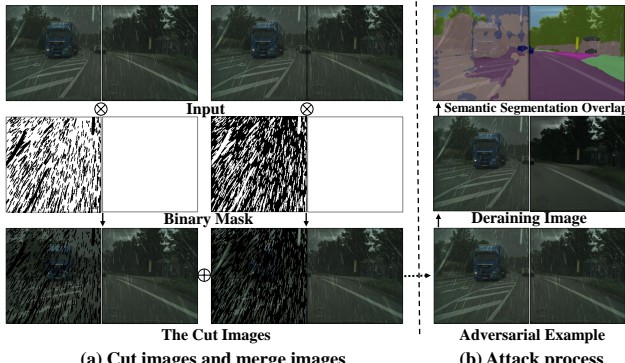

**(a) Cut images and merge images**   **(b) Attack process**

**Figure 1: A simple illustration of using a binary mask to cut an image and using the merged image for the attacking process. The left is from adversarial input/output, and the right is from the corresponding patch of clean input/output.**

substantial challenges to the accurate execution of these algorithms. Therefore, in real-world systems, it is a common practice to preprocess images captured under rainy conditions using a rain removal network. This preprocessing step is performed before feeding the images into downstream networks, to achieve the desired task performance under rainy weather conditions. Due to the fundamental role in computer vision and the extensive practical applications [4, 6, 13, 37, 39] of the semantic segmentation task, this paper focuses on it within an integrated system framework, where the rain removal network serves as the preprocessing module and the semantic segmentation network acts as the downstream task module.

Due to the powerful feature extraction ability of DNNs, both current rain removal methods [12, 50] and downstream task methods are predominantly based on deep learning. However, studies [15, 41] have demonstrated that subtle variations in images can alter the predictions of DNNs, despite these variations being incredibly subtle from the human visual system (HVS). As a result, computer vision tasks such as classification [5, 15, 31], detection [7, 24, 27, 46], and segmentation [1, 25, 36, 46, 49] have exhibited disastrous output results in the presence of well-designed adversarial examples. For the rain removal networks, Yu et al. [52] indicate that it is more vulnerable to adversarial attacks, as the small perturbations become less noticeable or detectable in rainy images.

However, the adversarial attack against an integrated system, such as the combination of the aforementioned networks, is highly challenging and remains unsolved. The reason is the substantial differences in network structure, learning objectives, data annotations, and training strategies between the two networks from different task hierarchies (low-level and high-level). As a result, common approaches for attacking multiple models, such as universal perturbations [19, 30] and ensemble attacks [18, 20], are ineffective in this scenario. When attacking a single model in isolation, the generated

adversarial examples often exhibit poor attack performance without considering the disparity across models designed for different tasks. If we attack both models, the challenge lies in aggregating the two sets of perturbations on the input rainy image. The intuitive way of directly adding both sets would disturb their respective distributions, consequently undermining the performance of each attack.

To address the aforementioned issues, we propose a novel attack mechanism with an image-centric perspective. This approach combines the two types of perturbations mentioned above in a non-overlapping manner. Specifically, we first propose the Remain Rain Attack (RRA), which neutralizes the effect of removing rain. In practice, adversarial attacks may succeed by modifying only a portion of the image pixels [40]. Inspired by this, since the RRA is only relevant to the rain regions, we constrain the generation of its perturbations within the rain region using a binary rain mask (see Fig. 1). Additionally, by leveraging the non-rain regions, we employ existing powerful adversarial attack algorithms [15, 23] to launch adversarial attacks on the semantic segmentation network. The finally fused perturbations from both attacks are then added to the input rainy image to generate the adversarial examples, forming the basis of our Cascaded Adversarial Attack (CAA). However, we observed that CAA exhibits instability when attacking models with robustness against adversarial attacks. This arises from the significant differences between the two generated perturbations. To overcome this issue, we propose two variants: Cascaded Adversarial Attack from Rain to Task (CAA-RT) and Cascaded Adversarial Attack from Task to Rain (CAA-TR). They incorporate the perturbations generated by one network into the optimization process of the other adversarial perturbation. The framework of the proposed attacks is illustrated in Fig. 2. It can be observed that our attacks preserve rain streaks and successfully mislead the output of the semantic segmentation network from Fig. 1.

Our key contributions are summarized as follows:

- We make the first attempt to attack an integrated system consisting of the rain removal network and the semantic segmentation network, which is widely used in real-world scenarios.
- We employ binary masks to combine diverse types of adversarial perturbations within a single image, which offers a unique perspective in tackling the challenge of simultaneously attacking models with substantial disparities. Additionally, we propose two variants to alleviate the notable differences between the two types of perturbations.
- Comprehensive experimental results demonstrate that our methods surpass attacks focused solely on a single network. This validation emphasizes the necessity of simultaneously considering all models and tasks in an integrated system.

## 2 RELATED WORK

### 2.1 Adversarial Attacks in Visual Tasks

Szegedy et al. [41] initially discovered the adversarial examples. Considering the work of Szegedy et al. [41] requires multiple optimizations, Goodfellow et al. [15] proposed the Fast Gradient Sign Method (FGSM). Kurakin et al. [23] introduced the Basic Iterative Method (BIM) by iteratively applying the FGSM with small step sizes. Subsequently, various stronger methods are proposed like the M-FGSM [9] and the Projected Gradient Descent (PGD) [29]. As the field of adversarial attacks advanced, extensive research has focused on attacking various computer vision tasks, such as classification [5, 15, 31] and object detection [7, 24, 27, 46], as well as semantic segmentation [1, 36, 46, 49].

Among various computer vision tasks, semantic segmentation enables a deep understanding of scenes by assigning meaningful labels to individual pixels, which is crucial for various practical applications. Early semantic segmentation methods relied on an encoder-decoder architecture [2, 28, 35]. Taking inspiration from the attention mechanism's ability to handle long-range dependencies in language processing [42], the introduction of non-local operations [43] into computer vision has resulted in numerous accurate models [11, 21]. In this paper, we focus on investigating the semantic segmentation task within the integrated system framework.

### 2.2 Adversarial Attacks in Rainy Scenes

Rain has been utilized in studies for adversarial attacks. Zhai et al. [54] treated rain streaks as adversarial noise and proposed a rain streak generation method that can simulate controllable factors such as rain intensity, direction, and brightness. By optimizing these factors, they generated rainy images that possess attack capabilities against DNNs. In addition to rain streaks, Liu et al. [26] introduced a novel method called AdvRD, which utilizes Generative Adversarial Network (GAN) techniques [14] to generate adversarial raindrop images. And similar works have been reported in [16].

In addition to rain-based attacks, researchers have explored the relationship between rain removal networks and adversarial attacks. Rain removal networks aim to eliminate rain streaks in rainy images, minimizing the disruptive effects caused by precipitation. While significant progress has been made in image restoration under rainy conditions [22, 34, 45, 48, 51, 57], the work of Yu et al. [52] has demonstrated the vulnerability of rain removal networks to adversarial attacks. Furthermore, they examined various types of adversarial attacks specific to rain removal problems and analyzed their effects on both human and machine vision tasks. However, previous adversarial attacks in rainy scenes focused on the rain removal network, neglecting the coexistence of the rain removal network and the downstream task network in practical systems. Currently, there is no existing work that simultaneously considers both aspects, and this work fills this gap.

## 3 METHODOLOGY

### 3.1 Problem Formulation

In the integrated system comprising the rain removal network and the semantic segmentation network, our attack objective is to deteriorate the output of the deraining network and semantic segmentation network by adding a small amount of visually unperceivable perturbations to the input rainy images. These perturbations are carefully designed to cause the failure of the final semantic segmentation task.

In the above scenario, we take a rainy image $X_{rain}$ as input, which follows the distribution $\mathbb{D}_{rain}$, and $\delta$ is an unperceivable perturbation, which is restricted in $\ell_p$-norm bound $\epsilon$. Thus, we get

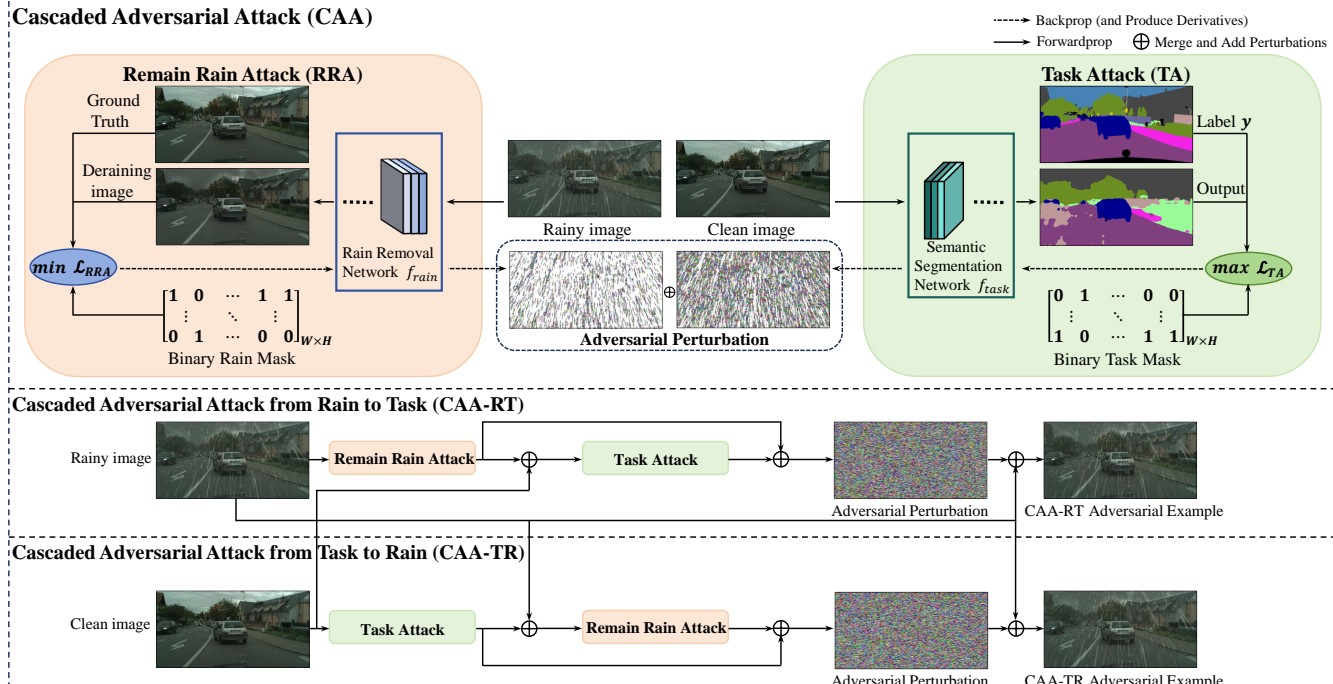

**Figure 2: The overview of the perturbation generation process for CAA (top row) and the adversarial example generation process for CAA-RT (second row) and CAA-TR (third row). In the Binary Rain Mask (BRM) of RRA, the value represented by 1 is the area where rain exists, while the value represented by 1 of the Binary Task Mask (BTM) in TA is the non-rain area, and the two are complementary. The perturbations obtained by segmenting RRA and TA with a binary mask can be injected into clean and rainy images, respectively, to generate the complementary perturbations required by CAA.**

the adversarial example $\hat{X}_{rain}$ by

$$\hat{X}_{rain} = X_{rain} + \delta$$
$$\|\delta\|_p \leq \epsilon. \tag{1}$$

We consider a rain removal network denoted as $f_{rain}$ and a semantic segmentation network denoted as $f_{task}$. The intermediate output result $M$ and $\hat{M}$ obtained after the rain removal operation can be represented in the following form:

$$M = f_{rain}(X_{rain}),$$
$$\hat{M} = f_{rain}(\hat{X}_{rain}). \tag{2}$$

The semantic segmentation output can be represented as follows:

$$X_{output} = f_{task}(M),$$
$$\hat{X}_{output} = f_{task}(\hat{M}). \tag{3}$$

It is worth noting that the accessibility of $M$ can vary across different systems and scenarios. $M$ can take various forms such as a deraining image, the output of the logits layer, or other representations. For instance, in an end-to-end system configuration, the preprocessing module and downstream task module are often integrated as a unified entity. In this paper, we do not delve into the specific form or accessibility of $M$, except for a brief demonstration in Table 6 to highlight the effectiveness of our methods when the deraining images are accessible. To ensure the generalization of our methods, this paper assumes that $M$ is inaccessible.

Here, we denote the measurement of the semantic segmentation effect as $Q(\cdot)$, and the goal of our attack is to degrade the semantic segmentation effect by minimizing $Q(\hat{X}_{output})$. Thus, the overall expectation of our adversarial attack algorithm is as follows:

$$\min_{\delta} \quad \mathbb{E}_{X_{rain} \sim \mathbb{D}_{rain}} \left[ Q(f_{task}(f_{rain}(X_{rain} + \delta))) \right]. \tag{4}$$

Unfortunately, finding an effective $\delta$ according to the equation (4) is a challenging task. As described in Section 1, it is difficult to deteriorate the semantic segmentation task by solely attacking $f_{rain}$ or $f_{task}$. When attacking solely the rain removal network, the optimization process lacks crucial information about the parameters and labels of the semantic segmentation network, making it difficult to generate perturbations that have a fatal impact on the semantic segmentation. On the other hand, perturbations act as noise, and the rain removal network can mitigate their effects through its own restoration capabilities [52] (see the supplementary materials for details). Moreover, the domain differences between rainy and clean images pose challenges when incorporating the perturbations generated by the semantic segmentation network into the input rainy image. Therefore, directly attacking the semantic segmentation network while disregarding the rain removal network in this system would result in poor attack performance (see Table 1). If we generate the adversarial perturbations separately for the deraining model and semantic segmentation model, combining these two perturbations on $X_{rain}$ is challenging. Simply adding them together

not only exceeds the given constraint conditions but also interferes with the well-optimized perturbation distributions for each other. To better accomplish the attack objectives, we propose two types of adversarial attack methods as follows.

## 3.2 Remain Rain Attack

In practice scenarios, rain streaks present significant interference to the downstream task (see Fig. 3). Motivated by this observation, we attack the rain removal network to undermine its rain removal ability. Specifically, we propose a method called Remain Rain Attack (RRA), which aims to make the rain streaks difficult to detect and remove, preserving a substantial amount of rain to disrupt the execution of the downstream task. We denote the perturbations generated by RRA as $\delta_{RRA}$. By adding these perturbations to the rainy image $X_{rain}$, the output $\hat{X}_{derain}$ is obtained through the $f_{rain}$, this operation can be formulated as:

$$\hat{X}_{derain} = f_{rain}(X_{rain} + \delta_{RRA}). \tag{5}$$

It is important to note that $\hat{X}_{derain}$ represents the complete output of the deraining network $f_{rain}$, rather than the intermediate product $\hat{M}$ embedded in the integrated system. Then, we define the objective function $\mathcal{L}_{RRA}$ to measure the distance between $\hat{X}_{derain}$ and $X_{rain}$:

$$\mathcal{L}_{RRA}(X_{rain}, \delta_{RRA}) = \left\| Mask \cdot \left| \hat{X}_{derain} - X_{rain} \right| - \mathbf{0} \right\|_2, \tag{6}$$

where the $\mathbf{0}$ is a zero matrix, which is the same size as the $\hat{X}_{derain}$. We utilize $\ell_2$ Euclidean distance to measure the difference between the input and output, which represents the level of rainwater retention. Here, the $Mask$ is a binary rain mask where the value 1 represents regions with rain, while the value 0 represents non-rain regions. In the RRA attack, our focus is solely on regions containing rain, disregarding non-rain regions. Therefore, incorporating $Mask$ in $\mathcal{L}_{RRA}$ enhances the computational efficiency and accuracy of $\mathcal{L}_{RRA}$ calculation. Our objective is to minimize $\mathcal{L}_{RRA}$, and the optimization of the objective function can be formalized as follows:

$$\min_{\delta_{RRA}} \quad \mathcal{L}_{RRA}(X_{rain}, \delta_{RRA}) \\ \text{s.t.} \quad \|\delta_{RRA}\|_p \leqslant \varepsilon. \tag{7}$$

We use the projected gradient descent (PGD) [29] to solve the optimization problem. To get the optimal $\delta_{RRA}$, we perform the following formula:

$$\delta_{RRA}^{t+1} = Proj(\delta_{RRA}^t - \alpha \text{sgn}(\nabla_{\delta_{RRA}^t} \mathcal{L}_{RRA}(X_{rain}, \delta_{RRA}))), \tag{8}$$

where $\nabla_{\delta_{RRA}^t} \mathcal{L}_{RRA}(X_{rain}, \delta_{RRA})$ is the gradient of the disrupting loss w.r.t $\delta_{RRA}$, $sgn$ extracts the sign of gradients, the term $\alpha$ controls the step length each iteration, $Proj(\cdot)$ denotes project the $\delta_{RRA}^t$ within the norm bound $(-\epsilon, \epsilon)$ and project the $X_{rain} + \delta_{RRA}$ within the valid space $(0, 1)$. The initial $\delta_{RRA}^0$ is sampled from the uniform distribution $U(-\epsilon, \epsilon)$, and the final adversarial perturbations $\delta_{RRA}^T$ is obtained after $T$ iterations.

## 3.3 Cascaded Adversarial Attack

The perturbations generated in Section 3.2 have demonstrated the ability to disrupt the execution of semantic segmentation tasks (see Table 1). However, as mentioned earlier, the RRA attack encounters challenges in posing a significant threat to the entire system due to

the exclusion of the semantic segmentation network from participating in the generation of adversarial perturbations. In fact, we have observed that powerful adversarial attack algorithms [5, 15, 23, 32] tend to generate adversarial perturbations that are concentrated in the main object regions of the image. This is because the target object is typically a primary factor influencing model decisions, and adversarial perturbations at the target region can more effectively disrupt the model's predictions [10]. This point is further illustrated by [38, 56], where the adversarial patch is used to occlude the main body of an object to deceive the task network. Inspired by this, we use $Mask$ to constrain the perturbations generated by RRA within the rain regions of the input rainy image. Specifically, for each iteration of $\delta_{RRA}^t$ in the iterative process, we multiply it by $Mask$. Consequently, to obtain $\delta_{RRA}^{t+1}$, the following operation is performed after applying equation (8):

$$\delta_{RRA}^{t+1} = \delta_{RRA}^t \cdot Mask. \tag{9}$$

Rain streaks occupy only a small portion of the rainy image and are uniformly distributed throughout the entire image. The main objects in the image mostly reside in non-rain regions. Therefore, for the majority of non-rain regions remaining in the input rainy image, we utilize them for attacking the semantic segmentation network. Regarding the attack on the downstream task network, existing attack algorithms have demonstrated strong performance [5, 15, 23, 32]. Here, we employ FGSM as an example, and we refer to the entire attack process on the semantic segmentation network as Task Attack (TA). For the input of TA, we represent it as $X_{clean}$, which follows the distribution $\mathbb{D}_{clean}$, and $X_{clean}$ is the ground truth of $X_{rain}$. We assume $y$ as the ground label of the semantic segmentation task, and the entire process of TA can be represented as follows:

$$\hat{X}_{clean} = X_{clean} + \epsilon \cdot sgn(\nabla_{X_{clean}} \mathcal{L}_{TA}(f_{task}(X_{clean}), y)), \tag{10}$$

where $\mathcal{L}_{TA}$ represents the cross-entropy loss, and our objective is to maximize $\mathcal{L}_{TA}$. The adversarial perturbation $\delta_{TA}$ can be obtained using the following equation:

$$\delta_{TA} = \epsilon \cdot sgn(\nabla_{X_{clean}} \mathcal{L}_{TA}(f_{task}(X_{clean}), y) \cdot (1 - Mask) \\ \|\delta_{TA}\|_p \leq \epsilon. \tag{11}$$

By splitting the rainy image with $Mask$, we obtain two non-overlapping perturbations, $\delta_{RRA}$ and $\delta_{TA}$. By adding $\delta_{RRA}$ and $\delta_{TA}$ to the input rainy image, we seamlessly integrate the generated perturbations from two models with significant disparities into a single image, which is referred to as Cascaded Adversarial Attack (CAA). This enables us to simultaneously attack both models without exceeding the constraint limit. Thus, the $\delta$ term in Equation (1) can be expressed as follows:

$$\hat{X}_{rain} = X_{rain} + \delta \\ = X_{rain} + \delta_{RRA} + \delta_{TA}. \tag{12}$$

Our experiments (see Table 1) indicate that directly combining and applying $\delta_{RRA}$ and $\delta_{TA}$ to the input rainy image may limit the individual effectiveness of each perturbation. This constraint arises from the notable disparities in the structures and parameters of the two models, as well as the distinct optimization objectives of RRA and TA. These two perturbations introduce various types and

(a) Visual comparison of the attacks conducted solely on a single network and our methods.

(b) Visual comparison between our proposed methods and the methods proposed by [52].

Figure 3: Qualitative comparison of attack methods on the integrated system consisting of MPRNet [53] and SSeg [58]. The first row represents the output obtained by MPRNet. The second row shows the semantic segmentation overlap of the final output results. The third row provides the difference between the prediction result and the ground truth. The implementation of TA in CAA is based on FGSM.

intensities of distortions in the image, which may interfere with each other, making the final perturbation effect unpredictable.

To address the above issue, we propose two variants based on the Cascaded Adversarial Attack framework: Cascaded Adversarial Attack from Rain to Task (CAA-RT) and Cascaded Adversarial Attack from Task to Rain (CAA-TR). They can smoothly combine the performance of RRA and TA. Specifically, CAA-RT performs RRA first and adds the perturbation $\delta_{RRA}$ onto the input $X_{clean}$ of TA, followed by the execution of TA to obtain $\delta_{TA}$. Finally, the adversarial example is obtained by utilizing Equation (12). Conversely, in CAA-TR, TA is applied first, and the generated perturbation $\delta_{TA}$ is added to $X_{rain}$. Subsequently, Equation (8) and (9) is employed to obtain $\delta_{RRA}$. Similar to CAA-RT, the final adversarial example is obtained using Equation (12). The generation formulas for $\delta_{RRA}$ in CAA-TR and $\delta_{TA}$ in CAA-RT are as follows:

$$\delta_{RRA}^{t+1} = Proj(\delta_{RRA}^t - \alpha\text{sgn}(\nabla_{\delta_{RRA}^t}\mathcal{L}_{RRA}(X_{rain}+\delta_{TA}, \delta_{RRA}))), \quad (13)$$

$$\delta_{TA} = \epsilon \cdot sgn(\nabla_{X_{clean}}\mathcal{L}_{TA}(f_{task}(X_{clean} + \delta_{RRA}), y) \cdot (1 - Mask). \quad (14)$$

Compared to CAA, CAA-TR performs the optimization of $\delta_{RRA}$ based on $\delta_{TA}$, allowing RRA to aggregate TA perturbation information and alleviate the distribution discrepancy between the two perturbations. The same principle applies to CAA-RT.

The pseudo-codes for our CAA-TF and CAA-RF algorithms are presented in the supplementary materials. In Fig. 2, we can see the process of our attack methods. During the inference stage, we begin by inputting the adversarial examples into the rain removal network. The output is then passed to the semantic segmentation network to obtain the final output.

## 4  EXPERIMENTS

### 4.1  Experimental Setups

**Datasets.** We utilize the Cityscapes [8] and RainCityscapes [17] datasets for semantic segmentation and rain removal tasks. The Cityscapes [8] is a widely recognized dataset for urban scene parsing, comprising 2975, 500, and 1525 images of size 2048 × 1024 for training, validation, and testing, respectively. RainCityscapes [17] is a synthetic deraining dataset generated based on Cityscapes [8], which consists of synthetic rainy images of size 512 × 256 and the rain-streaked images generated during the synthesis of rainy images. The rain removal model is more sensitive to image size due to the need for precise handling of image details and textures. Thus, we resized the images from Cityscapes [8] to a unified size of 512 × 256. In our experiment, we evaluated the effectiveness of attack methods on the validation set of Cityscapes [8] and RainCityscapes [17] datasets.

**Generation of Binary Masks.** To generate binary rain masks, we utilized the rain-streaked images provided by RainCityscapes [17]. Specifically, due to the high brightness of rain, we simply set pixels in the image that exceed a given threshold (0-255) to 1, while pixels below the threshold are set to 0. It is worth noting that in real-world scenarios, the generation of binary rain masks can be conveniently obtained by subtracting the deraining image from the input rainy image, as demonstrated in [52].

**Model Architectures.** [52] claims to provide a benchmark framework for adversarial attacks in rain removal scenarios. They combine modules from different rain removal networks to construct a

**Table 1: Results of various attack methods under different combinations of rain removal networks and semantic segmentation networks. The perturbations and random noises are restricted in $L_\infty$ norm bound 4/255. (·) represents the implementation of TA using FGSM or BIM. The best attack results are in boldface.**

| Methods | MPRNet [53] | | | | | | ARDNet [52] | | | | | |
|---|---|---|---|---|---|---|---|---|---|---|---|---|
| | SSeg [58] | | | PIDNet [47] | | | SSeg [58] | | | PIDNet [47] | | |
| Metrics | PSNR | SSIM | **mIoU** | PSNR | SSIM | **mIoU** | PSNR | SSIM | **mIoU** | PSNR | SSIM | **mIoU** |
| Clean rainy image | inf | 1 | 0.6262 | inf | 1 | 0.4985 | inf | 1 | 0.6740 | inf | 1 | 0.5385 |
| Random Noise | 36.56 | 0.930 | 0.6215 | 36.56 | 0.930 | 0.4966 | 36.56 | 0.930 | 0.5896 | 36.56 | 0.930 | 0.4714 |
| Input-close Attack [52] | 41.26 | 0.976 | 0.2654 | 41.26 | 0.976 | 0.1956 | 38.34 | 0.967 | 0.4667 | 38.34 | 0.967 | 0.3677 |
| Max-output Attack [52] | 37.71 | 0.958 | 0.3148 | 37.71 | 0.958 | 0.2205 | 37.85 | 0.968 | 0.5179 | 37.85 | 0.968 | 0.4035 |
| Down-stream Attack [52] | 38.35 | 0.958 | 0.3022 | 38.35 | 0.958 | 0.2257 | 38.62 | 0.959 | 0.3501 | 38.62 | 0.959 | 0.3068 |
| Down-stream Attack-v2 [52] | 39.60 | 0.968 | 0.2493 | 39.60 | 0.968 | 0.1858 | 39.11 | 0.965 | 0.4017 | 39.11 | 0.965 | 0.3382 |
| TA (FGSM) | 36.89 | 0.947 | 0.3365 | 36.86 | 0.949 | 0.1741 | 36.89 | 0.947 | 0.2935 | 36.86 | 0.949 | 0.3149 |
| TA (BIM) | 39.12 | 0.968 | 0.2591 | 39.07 | 0.968 | 0.1989 | 39.12 | 0.968 | 0.2466 | 39.07 | 0.968 | 0.2376 |
| RRA | 38.39 | 0.963 | 0.1600 | 38.39 | 0.963 | 0.1418 | 37.55 | 0.961 | 0.2845 | 37.55 | 0.961 | 0.2647 |
| CAA (FGSM) | 37.43 | 0.950 | 0.1311 | 36.89 | 0.950 | 0.1176 | 36.90 | 0.949 | 0.2856 | 37.42 | 0.950 | 0.2998 |
| CAA-RT (FGSM) | 37.44 | 0.949 | 0.1131 | 36.90 | 0.950 | 0.1132 | 36.90 | 0.949 | 0.2619 | 37.41 | 0.950 | 0.2613 |
| CAA-TR (FGSM) | 37.54 | 0.950 | **0.0752** | 36.48 | 0.945 | **0.1021** | 36.53 | 0.944 | 0.2416 | 37.52 | 0.951 | 0.2325 |
| CAA (BIM) | 38.56 | 0.964 | 0.1477 | 37.89 | 0.964 | 0.1223 | 37.90 | 0.963 | 0.2713 | 38.56 | 0.964 | 0.2600 |
| CAA-RT (BIM) | 38.58 | 0.964 | 0.1210 | 37.89 | 0.964 | 0.1153 | 37.92 | 0.963 | 0.2353 | 38.52 | 0.964 | 0.2338 |
| CAA-TR (BIM) | 38.65 | 0.964 | 0.1073 | 37.41 | 0.959 | 0.1116 | 37.53 | 0.960 | **0.2316** | 38.62 | 0.965 | **0.1994** |

rain removal network with strong robustness against adversarial attacks, which we refer to as Adversary Resistant Deraining Network (ARDNet). In line with their framework, we adopt the ARDNet [52] and SSeg [58] as our rain removal and semantic segmentation networks, respectively. To ensure the generality and practicality of our methods, we include MPRNet [53], which is widely utilized in real-world applications, and PIDNet [47], known for its capability in real-time semantic segmentation, to assess the effectiveness of our proposed methods.

**Evaluation Metrics.** We employ mean Intersection over Union (mIoU) as the evaluation metric for assessing the performance of semantic segmentation. A higher decrease in the mIoU value indicates better attack performance. We utilize widely-used quality metrics Peak Signal-to-Noise Ratio (PSNR) and Structural Similarity (SSIM) [44] to measure the concealment of perturbations. The values of PSNR and SSIM are computed by comparing the adversarial examples with clean rainy images.

**Attack Parameters.** In the paper, the TA mentioned in Section 3.2 was implemented using the FGSM [15] and BIM [23]. To ensure imperceptibility to the human eye, we set the $L_\infty$ norm bound $\epsilon$ as 4/255, $\alpha = \epsilon/4$. According to Fig. 4 and supplementary materials, iterations T for RRA and TA were set to 100 and 10, respectively. According to Table 3, unless otherwise specified, the threshold for the binary masks was set to 85.

## 4.2 Effectiveness of CAA

Fig. 4 shows the poor attack performance when using rainy images as inputs for TA, and the deraining images may not be accessible. Hence, we adopt clean images as inputs for TA. For attacking rain removal networks, [52] presents several approaches. The Max-output Attack aims to generate adversarial perturbations by maximizing the Euclidean norm between the input and output, with the objective of disrupting the image. The Input-close Attack generates

perturbations by minimizing the mean squared error loss between the input and output. These two approaches are commonly employed for attacking rain removal networks. Additionally, they introduce the Down-stream Attack, which utilizes Learned Perceptual Image Patch Similarity (LPIPS) [55] to degrade machine vision and impact downstream tasks. Furthermore, Down-stream Attack-v2 simultaneously leverages both human and machine vision to impact downstream tasks.

We systematically evaluated the effectiveness of various attack methods within the integrated system framework in Table 1. It is evident that attacking a single network alone falls short of posing a severe threat to the overall system. Simultaneously attacking both networks is necessary to significantly degrade the performance of the ultimate task. Despite the nearly complete success of FGSM and BIM in attacking semantic segmentation networks (see the supplementary materials), the underwhelming performance of TA can be attributed to two factors: the need to apply perturbations generated from clean images onto the input rainy images and the interference caused by the rain removal network with these perturbations. The methods proposed by [52] exhibit weak performance, particularly for ARDNet [52]. In contrast, our proposed CAA-RT and CAA-TR outperform any single-network attacks, posing a more lethal threat to the integrated system. Taking the system composed of MPRNet [53] and SSeg [58] as an example, CAA-TR (FGSM) outperforms the best single-network attack method (RRA) by 53%, and CAA-TR (BIM) outperforms it by approximately 32.9%. Similar improvements of approximately 27.99% and 21.29% are observed for the system composed of MPRNet and PIDNet [47], respectively. Comparable experimental results are also observed for ARDNet. Furthermore, as previously analyzed, the performance of CAA is influenced by the differences in perturbation distributions between RRA and TA. From the perspective of SSIM and PSNR, the changes

**Table 2: The impact of the components of CAA. Replace is short for the method employed for replacing the RRA.**

| Methods | Replace | input | PSNR | SSIM | mIoU |
|---------|---------|-------|------|------|------|
| CAA-RT | FGSM | Rainy image | 37.13 | 0.946 | 0.4494 |
|  | BIM | Rainy image | 37.12 | 0.946 | 0.4317 |
|  | FGSM | Clean image | 36.83 | 0.947 | 0.2959 |
|  | BIM | Clean image | 37.63 | 0.952 | 0.2806 |
|  | - | - | 36.90 | 0.949 | **0.2619** |
| CAA-TR | FGSM | Rainy image | 37.12 | 0.946 | 0.3437 |
|  | BIM | Rainy image | 37.11 | 0.946 | 0.3339 |
|  | FGSM | Clean image | 36.91 | 0.947 | 0.2927 |
|  | BIM | Clean image | 37.73 | 0.952 | 0.2614 |
|  | - | - | 36.53 | 0.944 | **0.2416** |

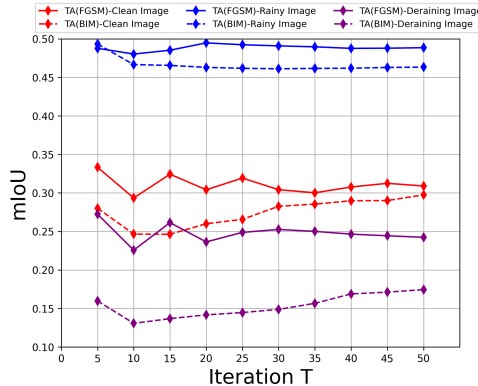

**Figure 4: The attack performance of TA under different inputs and iterations on SSeg [58] and ARDNet [52].**

in the images are nearly imperceptible to human vision (with PSNR values above 36.50 and SSIM values above 0.930).

Under the given experimental conditions, we observe that CAA-TR outperforms CAA-RT. This is primarily attributed to the fact that CAA incorporates the perturbation distribution from the previous attack in the subsequent attack. Consequently, the input of the subsequent attack is closer to the final adversarial example, enabling the adversarial perturbation generated by it to better align with the final adversarial example. This alignment enhances the effectiveness of achieving the subsequent attack objective. Additionally, due to the limited area of the rain region, RRA experiences a significant decrease in performance when constrained by a binary rain mask for perturbation generation. In contrast, TA continues to demonstrate excellent performance (see Table 4). As a result, CAA-TR enhances the effectiveness of RRA to a greater extent than CAA-RT enhances TA under such experimental conditions, leading to greater overall attack benefits.

To ensure fairness, we present qualitative visualizations of various attacks on MPRNet, where the attack methods proposed in [52] exhibit good performance. As shown in Fig. 3, our methods are capable of completely incapacitating the semantic segmentation network, while attacks on a single network still allow partial correct execution of the semantic segmentation task. Moreover, our methods also preserve a significant amount of rain in the output

**Table 3: Attacking results of CAA-TR and CAA-RT under different binary mask thresholds. Area(%) represents the percentage of the rain region to the total image area.**

| Methods | Threshold | Area(%) | PSNR | SSIM | mIoU |
|---------|-----------|---------|------|------|------|
| CAA-RT | 80 | 99.89% | 37.52 | 0.966 | 0.3191 |
|  | 85 | 32.90% | 36.90 | 0.949 | **0.2619** |
|  | 90 | 25.75% | 36.84 | 0.948 | 0.2731 |
|  | 95 | 22.57% | 36.85 | 0.948 | 0.2787 |
|  | 100 | 19.96% | 36.81 | 0.947 | 0.2742 |
| CAA-TR | 80 | 99.89% | 37.52 | 0.966 | 0.3056 |
|  | 85 | 32.90% | 36.53 | 0.944 | **0.2416** |
|  | 90 | 25.75% | 36.58 | 0.944 | 0.2501 |
|  | 95 | 22.57% | 36.60 | 0.945 | 0.2520 |
|  | 100 | 19.96% | 36.63 | 0.945 | 0.2594 |

**Table 4: The impact of binary masks on RRA and TA.**

| Methods | Settings | PSNR | SSIM | mIoU |
|---------|----------|------|------|------|
| RRA | w/ BRM | 40.94 | 0.988 | 0.3813 |
|  | w/ BTM | 39.39 | 0.966 | 0.4428 |
|  | w/ final BRM | 41.45 | 0.989 | 0.5715 |
|  | w/ final BTM | 40.14 | 0.970 | 0.5270 |
|  | w/o BRM | 37.55 | 0.961 | **0.2845** |
| TA | w/ BRM | 40.65 | 0.985 | 0.5545 |
|  | w/ BTM | 39.46 | 0.960 | 0.3036 |
|  | w/ final BRM | 40.80 | 0.985 | 0.6177 |
|  | w/ final BTM | 39.50 | 0.961 | 0.4074 |
|  | w/o BTM | 36.88 | 0.947 | **0.2935** |

images, indicating the simultaneous attack on both the rain removal network and the semantic segmentation network. From Fig. 3, our proposed RRA preserves more rain streaks (particularly rain texture details) compared to the Input-close Attack.

### 4.3 Ablation Study

Table 1 and supplementary materials demonstrate the superiority of SSeg [58] over PIDNet [47] on the Cityscapes [8] dataset with an image size of $512 \times 256$. Additionally, FGSM is a commonly used method due to its speed and convenience, and the application of visual algorithms in real-world scenarios always considers security. For brevity and without sacrificing generality, we conducted the ablation experiments using FGSM-based TA on SSeg and ARDNet.

**Impact of RRA on CAA.** We examined the influence of different components in CAA by substituting RRA with TA. Specifically, we used TA to generate perturbations in the rain region, as indicated in Table 2. In this scenario, CAA degenerates into an attack solely on the semantic segmentation network. When rainy images were used as inputs for TA, the attack performance of CAA-RT and CAA-TR significantly declined. This decline in performance can be attributed to the limited capability of the semantic segmentation network to handle rainy images. Consequently, the perturbations generated by TA within the rain region become ineffective (see Fig. 4). To ensure fairness, we conducted experiments using clean

**Table 5: CAA Transferability. For brevity, we only provide the mIoU metric. The contents of each cell are CAA-TR / CAA-RT.**

| Source Model | Target Model | | | |
|---|---|---|---|---|
| | MPRNet+SSeg | MPRNet+PIDNet | ARDNet+SSeg | ARDNet+PIDNet |
| **MPRNet+SSeg** | **0.0752 / 0.1131** | 0.2482 / 0.2747 | 0.3480 / 0.3522 | 0.5928 / 0.5925 |
| **MPRNet+PIDNet** | 0.1491 / 0.2102 | **0.1021 / 0.1132** | 0.4532 / 0.4594 | 0.2513 / 0.2548 |
| **ARDNet+SSeg** | 0.3155 / 0.3145 | 0.5779 / 0.5799 | **0.2416 / 0.2619** | 0.4697 / 0.5133 |
| **ARDNet+PIDNet** | 0.4297 / 0.4174 | 0.3448 / 0.3336 | 0.3679 / 0.4036 | **0.2325 / 0.2613** |

**Table 6: Attacking results on SSeg [58] and ARDNet [52] in the presence of accessible deraining images.**

| Methods | PSNR | SSIM | mIoU |
|---|---|---|---|
| TA | 36.86 | 0.947 | 0.2258 |
| CAA | 36.92 | 0.949 | 0.2236 |
| CAA-RT | 38.00 | 0.964 | **0.1492** |
| CAA-TR | 36.56 | 0.945 | 0.1861 |

images as inputs for TA as well. However, the disruption caused by the binary mask resulted in poorer attack performance compared to attacks that solely attacked the semantic segmentation network. Therefore, this experiment demonstrates that the binary mask acts as a binding factor, which combines two distinct types of adversarial attack methods. The effectiveness of CAA lies in its ability to simultaneously attack both networks.

**Influence of Binary Masks on RRA and TA.** Table 4 illustrates that restricting the attack perturbations to a specific region results in decreased attack performance. Furthermore, incorporating a binary mask during the iterative perturbation generation process is significantly more effective than adding it after perturbation optimization is completed. TA exhibits greater robustness to binary masks (decreased by approximately 3.44%), while RRA is more sensitive to them (decreased by approximately 34.02%). This is primarily due to TA occupying a larger portion of the image (see Table 3). When we exchange the mask regions for each attack method, we observe a significant decrease in the performance of TA (decreased by approximately 88.92%). In this scenario, where TA occupies regions with a significant amount of rain, it also affects the performance of TA. Moreover, when RRA occupies a significant portion of the image without rain, the results are unsatisfactory. This highlights the importance of focusing on the accuracy of the rain region in the attack strategy. We also present the attack results of CAA-RT and CAA-TR under different binary mask thresholds ranging from 80 to 100 with a step size of 5 (see Table 3). It can be observed that our methods perform well under different mask thresholds. When the threshold is set to 80, CAA-RT and CAA-TR almost degenerate into RRA.

## 4.4 Transferability of CAA-TR and CAA-RT

We explore the transferability of CAA-TR and CAA-RT across systems composed of different rain removal and semantic segmentation networks, and Table 5 shows the results. We find that the CAA-TR and CAA-RT exhibit favorable transferability across different systems, as long as these systems include the source model that

generates adversarial examples. Achieving comprehensive defense against CAA-TR and CAA-RT necessitates replacing all networks within the integrated system, incurring significant defense costs. Furthermore, the robustness of ARDNet against adversarial attacks hampers the transferability of our methods. This underscores the importance for system designers to consider robustness against adversarial attacks in practical application scenarios.

## 4.5 Accessible Deraining Images

As mentioned earlier, deraining images may not be accessible. In this section, we assume access to deraining images. From Fig. 4, when deraining images are used as inputs for TA, the performance is improved. This is due to deraining images being closer to the rainy images compared to clean images, resulting in perturbations of TA that better align with the distribution of rainy images. We implement CAA-RT and CAA-TR using deraining images as inputs of FGSM-based TA, as depicted in Table 6. CAA-RT and CAA-TR remain effective when deraining images are accessible, exhibiting even stronger performance with an improvement of approximately 33.92% and 17.58% over TA, respectively. This highlights the universality of our methods. It is crucial to highlight that in this scenario, CAA-RT surpasses CAA-TR, benefiting from the greatly improved performance of TA facilitated by the utilization of deraining images. Consequently, the more effectively CAA leverages the performance of TA, the more it is amplified.

## 5 CONCLUSION

This work presents the first exploration to simultaneously attack both low-level rain removal networks and high-level computer vision networks, exposing the vulnerability of such widely deployed systems in real-world scenarios. Specifically, we propose the Cascaded adversarial attack (CAA), which innovatively utilizes binary masks to partition the image into rain and non-rain regions. One region is utilized to attack the rain removal network, while the other region attacks the downstream task network. By concentrating two distinct sets of perturbations on a single rainy image, even after undergoing the preprocessing operations of the rain removal network, the downstream task is severely affected. We also propose two variants of CAA, namely CAA-RT, and CAA-TR, to minimize the divergence between the two generated perturbations. Extensive experiments demonstrate the superiority of our approaches over methods that solely attack a single network. Moreover, as long as a system has a source model of adversarial examples, our methods can attack this system. In the future, we will investigate approaches to perform CAA in black-box settings.

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
