# OpenReview forum: "Cascaded Adversarial Attack: Simultaneously Fooling Rain Removal and Semantic Segmentation Networks"
_acmmm.org/ACMMM/2024/Conference — MM2024 Poster_

### Official Review · Reviewer_CJzs · 2024-05-25

**Rating:** 3
**Confidence:** 4

**Summary:**

This work introduces a novel adversarial attack method aimed at simultaneously targeting both rain removal and semantic segmentation networks within an integrated system. The approach, named Cascaded Adversarial Attack (CAA), employs binary masks to efficiently focus different types of perturbations on specific regions of an input image, effectively degrading the performance of downstream tasks. The paper highlights the method's ability to undermine the typical preprocessing of rain images before they undergo further analysis, which is a common practice in applications.

**Strengths:**

1. The paper excels in introducing a novel Cascaded Adversarial Attack (CAA) that simultaneously targets rain removal and semantic segmentation networks, demonstrating effective dual-targeting of integrated systems.
2. It utilizes binary masks for precise perturbation control, effectively enhancing attack efficiency.
3. Extensive evaluations confirm the superiority of this approach over traditional single-network attacks.

**Limitations:**

1. Limited Literature Review: The discussion of related work, especially on approaches that simultaneously handle upstream rain removal and downstream semantic segmentation, seems sparse.
2. Lack of Comparative Analysis: The authors do not compare their method with existing attacks specifically designed for segmentation models.
3. Insufficient Performance Evaluation: The performance evaluation of the proposed method lacks comprehensiveness; it does not consider varying attack intensities or assess the method across different datasets, which are critical for understanding the applicability of the attack across different scenarios.

[1] PEARL: Preprocessing enhanced adversarial robust learning of Image deraining for semantic segmentation. (ACM MM 2023)
[2] Segpgd: An effective and efficient adversarial attack for evaluating and boosting segmentation robustness. (ECCV 2022)

**Suitability:**

2

---

### Official Review · Reviewer_pupS · 2024-05-25

**Rating:** 4
**Confidence:** 2

**Summary:**

This paper introduce a new setting for adversarial attack, called cascaded adversarial attack, where the goal is attack two cascaded tasks such as rain removal and semantic segmentation.

**Strengths:**

1. The proposed cascaded setting is interesting. Different task orders are studied.

2. The experiments is relatively comprehensive and show the effectiveness of the method.

3. The writing is good in general with nice illustrations.

**Limitations:**

1. Although the cascaded setting is interesting, the targeted tasks are too specific (rain removal and semantic segmentation). Ideally, this cascaded adversarial attack should be studied as a general problem where the tasks can be arbitrary. The experiments should also show the applicability of the method to different combinations of tasks. Therefore, the scope of this paper is limited.

2. As for the method, it is also specific to the rain removal and semantic segmentation, which might not be easily applicable to other combinations of tasks.

3. It will be better to use less abbreviations. The 'cut image' in figure 1 is not an accurate saying, it is not a 'cut'.

**Suitability:**

2

---

### Official Review · Reviewer_dRK3 · 2024-05-27

**Rating:** 4
**Confidence:** 3

**Summary:**

This paper aims to simultaneously attack both low-level rain removal networks and high-level computer vision networks, exposing the vulnerability of such widely deployed systems in real-world scenarios. Cascaded adversarial attack (CAA)  utilizes binary masks to partition the image into rain and non-rain regions. One region is utilized to attack the rain removal network, while the other region attacks the downstream task network.

**Strengths:**

1 The proposed method is able to simultaneously attack both low-level rain removal networks and high-level computer vision networks.
2 CAA and its two variants CAA-RT , CAA-TR are proposed to to minimize the divergence between the two generated perturbations.

**Limitations:**

1 The motivation of designing methods for fooling rain removal methods is not clear.
2 It seems that the proposed method only can be used in the digital world.

**Suitability:**

2

---

### Official Review · Reviewer_4cQ1 · 2024-05-31

**Rating:** 4
**Confidence:** 4

**Summary:**

This paper introduces a novel approach called "Cascaded Adversarial Attack" (CAA), which simultaneously targets de-raining networks and semantic segmentation networks within an integrated system. By using binary masks to split the input image into two parts, the CAA allows different distributions of disturbances to concentrate on the same input image, thereby affecting both networks simultaneously. Additionally, the paper proposes two variants of CAA that integrate the generation of the two disturbances during the optimization process, reducing the differences between them and enhancing attack performance.

**Strengths:**

(1) This is the first attempt to attack an integrated system that targets networks for both de-raining and semantic segmentation tasks. The method innovatively uses binary masks to effectively combine two types of adversarial perturbations, offering a unique perspective for tackling the challenge of attacking models with significant structural and objective differences.
(2) Extensive experimental results validate the effectiveness of the method.
(3) The figures are well-drawn.

**Limitations:**

The method proposed in the paper includes several steps. To further enhance the practical value and completeness of the paper, I suggest that the author add the time complexity analysis of the proposed algorithm.
Missing references:
UniParser: Multi-Human Parsing with Unified Correlation Representation Learning
Semantic Compression Embedding for Generative Zero-Shot Learning.
Understanding humans in crowded scenes: Deep nested adversarial learning and a new benchmark for multi-human parsing

**Suitability:**

3

---

### Meta-Review · Area_Chair_HZ45 · 2024-07-03

**Recommendation:** Accept (Poster)
**Confidence:** 5

**Metareview:**

This paper was reviewed by four experts in the field. The recommendations are (Borderline Accept x 3, Borderline Reject). Based on the reviewers' feedback, the decision is to recommend the acceptance of the paper. The reviewers did raise some valuable concerns (especially detailed complexity analysis, more comparisons with missing related works, more detailed experimental ablations, and polished writing and presentations, etc.) that should be addressed in the final camera-ready version of the paper. The authors are encouraged to make the necessary changes to the best of their ability. We congratulate the authors on the acceptance of their paper.